# Adaptive Evolution for the Efficient Production of High-Quality d-Lactic Acid Using Engineered *Klebsiella pneumoniae*

**DOI:** 10.3390/microorganisms12061167

**Published:** 2024-06-08

**Authors:** Bo Jiang, Jiezheng Liu, Jingnan Wang, Guang Zhao, Zhe Zhao

**Affiliations:** 1State Key Laboratory of Microbial Technology and Institute of Microbial Technology, Shandong University, Qingdao 266237, China; jbo0423@163.com (B.J.); liujz@qibebt.ac.cn (J.L.); wangjingnan190@163.com (J.W.); 2CAS Key Laboratory of Biobased Materials, Qingdao Institute of Bioenergy and Bioprocess Technology, Chinese Academy of Sciences, Qingdao 266101, China; 3Shandong Energy Institute, Qingdao 266101, China; 4Qingdao New Energy Shandong Laboratory, Qingdao 266101, China; 5University of Chinese Academy of Sciences, Beijing 100049, China

**Keywords:** d-lactic acid, adaptive evolution, organic acid tolerance, stress response, *Klebsiella pneumoniae*

## Abstract

d-Lactic acid serves as a pivotal platform chemical in the production of poly d-lactic acid (PDLA) and other value-added products. This compound can be synthesized by certain bacteria, including *Klebsiella pneumoniae*. However, industrial-scale lactic acid production in *Klebsiella pneumoniae* faces challenges due to growth inhibition caused by lactic acid stress, which acts as a bottleneck in commercial microbial fermentation processes. To address this, we employed a combination of evolutionary and genetic engineering approaches to create an improved *Klebsiella pneumoniae* strain with enhanced lactic acid tolerance and production. In flask fermentation experiments, the engineered strain achieved an impressive accumulation of 19.56 g/L d-lactic acid, representing the highest production yield observed in *Klebsiella pneumoniae* to date. Consequently, this strain holds significant promise for applications in industrial bioprocessing. Notably, our genome sequencing and experimental analyses revealed a novel correlation between UTP-glucose-1-phosphate uridylyltransferase GalU and lactic acid resistance in *Klebsiella pneumoniae*. Further research is warranted to explore the potential of targeting GalU for enhancing d-lactic acid production.

## 1. Introduction

Lactic acid is a significant chiral chemical that finds widespread use in several industrial chemicals such as food additives, pharmaceuticals, fine chemicals, and biopolymers [1]. Recently, there has been a growing emphasis on environmental conservation, leading to increased recognition of polylactic acid (PLA) as a sustainable source of plastic material [2]. However, the production of PLA necessitates the utilization of monomeric lactic acid that possesses exceptional quality, characterized by both high optical and chemical purity [3,4,5]. Different monomers can undergo polymerization and form distinct types of bioplastics, including poly l-lactic acid (PLLA), poly d-lactic acid (PDLA), and poly d,l-lactic acid (PDLLA). The process of fermenting l-lactic acid is widely recognized and extensively studied in academic circles. d-lactic acid has gained significant importance recently and has found extensive applications in various sectors, such as packaging, coatings, textiles, and the automotive industry [4].

d-lactic acid can be generated through fermentation using various microbial strains, including wild-type *Lactobacillus* and *Sporolactobacillus*, as well as metabolic-engineered strains from the *Escherichia*, *Saccharomyces*, and *Klebsiella* genera [6]. *Klebsiella pneumonia*, with its rapid growth rate in minimal medium and substrate versatility (including glycerol and monosaccharides), has been a focus of research [6]. It is already widely used as a microbial factory for the production of 3-hydroxypropionate [7], 1,3-propanediol [8], hyaluronic acid [9], 2-hydroxyisovalerate [10], and d-lactic acid [6,11]. Notably, the d-lactic acid dehydrogenase of the *K. pneumoniae* strain exhibits excellent selectivity, and *K. pneumoniae* ATCC25955 is able to produce optically pure d-lactate with no detectable l-lactate in culture [11]. In our previous study [6,11], we engineered *K. pneumoniae* ATCC25955 to produce optically pure d-lactic acid with high productivity and yield from glycerol and glucose, respectively. However, there remains a scarcity of reports on metabolic engineering strategies for d-lactic acid production in *K. pneumoniae* [6,11,12]. Investigating *K. pneumoniae* further may reveal novel approaches to enhance d-lactic acid production and broaden its industrial applications.

The main barrier to commercial lactic acid production via microbial fermentation is growth inhibition caused by various environmental stresses encountered during industrial bioprocesses [13]. Among these, the growth limitation caused by the fermentation product (lactic acid) is noteworthy. The toxicity of organic acids results from the combined effects of anion and acidic pH [14]. Specifically, undissociated lactic acid concentration in the medium rises in tandem with bacterial growth and lactate accumulation [15]. The membrane-soluble undissociated form of lactic acid diffuses into the cytoplasm, leading to rapid dissociation, proton release, and anion release [16]. If the proton accumulation rate exceeds the cytoplasmic buffering capacity and efflux system capabilities, cellular functioning is impaired [15,17]. In this process, the exporting of protons and acid anions via H^+^-ATPase and efflux pumps, respectively, consumes a significant amount of ATP [18]. Notably, intracellular accumulation of acid anions may be of greater importance than proton release in inhibiting cell growth [15]. Acid anion accumulation within the cell increases cellular osmolarity, causing lethal turgor pressure [19]. It may also induce a direct feedback inhibition of important metabolic pathways [19]. For example, Roe et al. [20] proved that methionine production was specifically suppressed by acetate accumulation. Hence, improving bacterial tolerance to low pH and lactate is critical for achieving balanced cell growth and a high lactic acid titer.

Microbial adaptive laboratory evolution (ALE) stands out as a simple yet effective technique for optimizing industrial strains that produce high-value metabolic products. Additionally, ALE provides insights into microbial resistance to various stressors. For example, in a previous study, we acquired phloroglucinol-tolerant *Escherichia coli* strains via ALE, and three mutations (Δ*sodB*, Δ*clpX*, and *fetAB* overexpression) proved of great assistance in the tolerance improvement [21]. Mazzoli et al. [22] increased the lactate tolerance of *C. thermocellum* strains LL345 (parent strain) and LL1111 (Δ*adhE*, lactic acid overproducing strain) through ALE. Svetlitchnyi et al. [23] obtained evolved strains originating from *Caldicellulosiruptor* sp. DIB 104C by ALE, which could produce unusually large amounts of l-lactic acid from microcrystalline cellulose in fermenters. Tian et al. [2] employed adaptive evolution to breed and select a high-performance *Lactobacillus paracasei* strain that could produce 221.0 g/L of l-lactic acid in open fermentation with a high initial glucose concentration.

In this study, we conducted ALE to generate lactic acid-tolerant strains. These strains were cultivated in media with gradually increasing lactic acid levels, all without pH adjustment. Our subsequent genome resequencing analysis provided insights into adaptation mechanisms and potential strategies for overcoming bottlenecks under selective pressure. Eight evolved strains were subjected to genome sequencing, revealing four single nucleotide polymorphism (SNP) changes. Additionally, through transcriptional analysis and colony-forming unit (CFU) ratio testing, we discovered a novel correlation between the *galU* gene and bacterial lactic acid resistance. Notably, this finding represents the first time such a relationship has been observed. In addition, as the Δ*budB*Δ*ackA*Δ*adhE* strain Q2702 obtained a high production and yield of d-lactic acid in our previous study [6], we deleted *ackA*, *adhE,* and *budB* genes in our evolved strains. The engineered strains performed better in glucose-containing media, which have higher lactic acid tolerance, growing faster and producing more d-lactic acid than the control strain Q2702. In particular, Q5224 has superior growth ability and can produce 19.56 g/L of lactic acid in shake flask fermentation, which was almost 55.5% higher than that of Q2702, making it a promising candidate for industrial d-lactic acid production.

## 2. Materials and Methods

### 2.1. Bacterial Strains and Growth Conditions

All strains were listed in Table 1, and all primers were listed in Appendix A. Bacteria were cultured at 37 °C in Luria-Bertani broth or the fermentation medium. The fermentation medium consisted of the following components per liter: glucose (20 g); NH_4_Cl (5.4 g); KH_2_PO_4_ (2 g); K_2_HPO_4_ (1.6 g); citric acid (0.42 g); MgSO_4_ ·7H_2_O (0.2 g); and trace elements stock solution (1 mL). The trace element solution includes Na_2_MoO_4_ ·2H_2_O (0.005 g/L), H_3_BO_3_ (0.062 g/L), CuCl_2_ ·2H_2_O (0.17 g/L), CoCl_2_ ·6H_2_O (0.476 g/L), ZnCl_2_ (0.684 g/L), MnCl_2_ ·4H_2_O (2 g/L), FeCl_3_·6H_2_O (5 g/L), and concentrated hydrogen chloride (HCl) (10 mL/L). Antibiotics were added when necessary at final concentrations of 50 μg/mL for chloramphenicol. *E. coli* DH5α was used as a host for the preparation of plasmid DNA, and *E. coli* χ7213 was used for the preparation of suicide plasmids. Diaminopimelic acid (DAP) at 50 μg/mL supported the growth of χ7213 strain. Additionally, LB agar containing 10% sucrose facilitated *sacB* gene-based counter-selection in allelic exchange experiments.

In this study, plasmids were constructed by digesting PCR fragments containing the target gene and cloning them into corresponding vectors as normal. *Klebsiella pneumonia* ∆*rcsA*, *Klebsiella pneumonia* ∆*galU*, *Klebsiella pneumonia* ∆*ydhS*, *Klebsiella pneumonia* A1∆*adhE*∆*ackA*∆*budB*, *Klebsiella pneumonia* A4∆*adhE*∆*ackA*∆*budB*, *Klebsiella pneumonia* B1∆*adhE*∆*ackA*∆*budB* was constructed by homologous recombination using suicide plasmids [24].
microorganisms-12-01167-t001_Table 1Table 1Strain and plasmid used in this study.Strain and PlasmidDescriptionSourceStrains

*E. coli* DH5αF^−^
*supE*44 Δ*lacU*169 (*ϕ*80 *lacZ* Δ*M15*) *hsdR*17 *recA*1 *endA1 gyrA*96 *thi*-1 *relA*1lab collection*E. coli* χ7213*thi-1 thr-1 leuB6 glnV44 fhuA21 lacY1 recA1 RP4-2-Tc*::Mu λ*pir* Δ*asdA4* Δ*zhf-2*::Tn10[25]*E.coli* CC118 (λpir+)*araD*139 Δ(*ara, leu*)7697 Δ*lacX*74 *phoA*Δ20 *galE galK thi rpsE rpoB argEam recA*1University of Pennsylvania. Dieter M. SchifferliQ1188*K. pneumoniae* ATCC25955ATCCQ4441*K. pneumoniae* ATCC25955 evolved strain A1this studyQ4442*K. pneumoniae* ATCC25955 evolved strain A2this studyQ4443*K. pneumoniae* ATCC25955 evolved strain A3this studyQ4444*K. pneumoniae* ATCC25955 evolved strain A4this studyQ4445*K. pneumoniae* ATCC25955 evolved strain A8this studyQ4446*K. pneumoniae* ATCC25955 evolved strain B1this studyQ4447*K. pneumoniae* ATCC25955 evolved strain C8this studyQ4448*K. pneumoniae* ATCC25955 evolved strain D3this studyQ4589*K. pneumoniae* ATCC25955 Δ*rcsA*this studyQ4590*K. pneumoniae* ATCC25955 1780734 SNP mutation (A<->G)this studyQ4607*K. pneumoniae* ATCC25955 Δ*galU*this studyQ4608*K. pneumoniae* ATCC25955 Δ*ydhS*this studyQ4616*K. pneumoniae* ATCC25955 Δ*galU*/pACYCDuet1-Plac_1-6_lacOthis studyQ4601*K. pneumoniae* ATCC25955/pACYCDuet1-Plac_1-6_lacO-*galU*this studyQ4609*K. pneumoniae* ATCC25955Δ*galU*/pACYCDuet1-Plac_1-6_lacO-*galU*this studyQ2702*K. pneumoniae* ATCC25955 Δ*budB* Δ*ackA* Δ*adhE*this studyQ5221*K. pneumoniae* ATCC25955 A1 Δ*budB* Δ*ackA* Δ*adhE*
this studyQ5224*K. pneumoniae* ATCC25955 A4 Δ*budB* Δ*ackA* Δ*adhE*this studyQ5227*K. pneumoniae* ATCC25955 B1 Δ*budB* Δ*ackA* Δ*adhE*this studyPlasmids

pACYCDuet1-Plac_1-6_lacOrep_p15A_ Cm^R^ *lacI* Plac_1-6_lab collectionpRE112*oriT oriV sacB* Cm^R^lab collection


For shake flask cultivation, the strains were cultured in a 250-mL flask containing 100 mL of medium at 37 °C in an orbital incubator shaker running at 180 rpm. After 72 h, samples were withdrawn to determine the cell mass, glucose, d-lactic acid, and by-products. To maintain pH stability, 25% NH_3_·H_2_O was added every 12 h (with the pH dropping to approximately 4). Glucose levels were measured using an M-100 biosensor analyzer (Shenzhen Sieman Technology Co., Ltd., Shenzhen, China), and an additional 20 g/L glucose was added when the initial carbon source was nearly exhausted. All shaking experiments were carried out in triplicate.

### 2.2. Susceptibility Assay of Lactic Acid and Hydrogen Chloride

The tolerance assays of lactic acid were conducted in the fermentation medium containing 0, 0.5, 1.0, 2.0, or 5.0 g/L lactic acid. The wild type was cultured overnight and re-inoculated (1:50) into 100 mL of the fermentation medium in a 250 mL flask. Lactic acid susceptibility was monitored by the measurement of OD_600_ and CFU.

The survival assays of lactic acid were conducted in the fermentation medium containing 0 or 3.0 g/L lactic acid. Bacterial cells were cultured overnight, re-inoculated (2:100) in the fermentation medium and grown to an OD_600_ of 1.8. Indicated lactic acid was added into the fermentation medium, and strains were grown for another several hours before the cells were collected to determine the CFU.

The survival assays of hydrogen chloride were conducted in the fermentation medium with or without the hydrogen chloride challenge. Bacterial cells were cultured overnight, re-inoculated (2:100) in the fermentation medium and grown to an OD_600_ of 1.8. Indicated hydrogen chloride was added into the fermentation medium, and strains were grown for another several hours before the cells were collected to determine the CFU. Survival (%) = (CFU with lactic acid challenge/CFU without lactic acid challenge) × 100%.

### 2.3. Adaptive Evolution of K. pneumoniae Strain

Adaptive evolution of *K. pneumoniae* strain was performed. The strain was cultivated in the fermentation medium supplemented with 3 g/L yeast extract, reaching an optical density at 600 nm (OD_600_) of 1.8. It was then challenged with 2 g/L lactic acid for 10 h before being recovered in LB broth overnight. The resulting LB culture was used as the inoculum for the subsequent selection round with higher lactic acid concentration. To isolate evolved mutants, culture was spread onto LB agar plates, and single colonies were subjected to LA tolerance assay. Eight colonies with the highest lactic acid tolerance were designated as A1, A2, A3, A4, A8, B1, C8, and D3, and their genomes were resequenced by the Beijing Genomics Institution (BGI).

### 2.4. Quantitative Real-Time PCR

Bacterial cells were cultured overnight, re-inoculated (2:100) in the fermentation medium and grown to an OD_600_ of 1.8. Then, 3 g/L lactic acid was added into the medium, and strains were grown for another 4 h. The cells were collected before or after the addition of lactic acid.

Quantitative real-time PCR (qRT-PCR) was performed as previously described [26]. Total RNA was extracted from bacterial culture using the EASYSpin Plus bacterial RNA quick extract kit (Aidlab Biotechnologies, Beijing, China). RNA concentration was determined by spectrophotometry at 260 nm. Genomic DNA removal and cDNA synthesis were performed using the PrimeScript RT reagent Kit with gDNA Eraser (Takara, Shiga, Japan). qRT-PCR was conducted using TB Green Premix Ex Taq (Takara, Shiga, Japan) on the QuantStudio 1 system (Applied Biosystems, Waltham, MA, USA). Constitutively transcribed gene *rpoD* served as a reference control for normalizing the total RNA quantity of different samples. Relative differences in mRNA levels were calculated using the ΔΔCt method [27].

### 2.5. Analysis of d-Lactic Acid Production

Biomass was monitored using a UV visible spectroscopy system (Varian Cary 50 Bio, Palo Alto, CA, USA) at 600 nm. The fermentation products of d-lactic acid, acetate, ethanol, and succinate were detected by HPLC with a refractive index detector (RI-150, Thermo Spectra System, Waltham, MA, USA) and ion exchange column (Aminex^®^ HPX-87H, 7.8 × 300 mm, BioRad, Hercules, CA, USA) at 60 °C. The mobile phase consisted of 5 mM H_2_SO_4_ as the mobile phase, and the flow rate was set at 0.5 mL/min [6].

## 3. Results

### 3.1. ALE of K. pneumoniae in a High Concentration of Lactic Acid

We employed ALE to bolster lactic acid tolerance in d-lactic acid-producing *K. pneumoniae*. Initially, we investigated the cell growth of the wild-type *K. pneumoniae* (ATCC25955) in a fermentation medium (Figure 1A). As lactic acid concentrations increased, the proliferation and survival of the wild-type strain gradually diminished (Figure 1B,C). Notably, the colony-forming unit (CFU) analysis revealed that the bacteria had limited survival capacity at a lactic acid concentration of 2.0 g/L (Figure 1C). To improve the lactic acid tolerance of *K. pneumoniae*, we conducted an adaptation experiment by gradually increasing lactic acid concentrations in the growth medium from 2.0 to 5.1 g/L without pH adjustment (Figure 2A). Subsequently, we isolated eight evolved strains, designated A1, A2, A3, A4, A8, B1, C8, and D3. Remarkably, these evolved strains exhibited significantly improved lactic acid tolerance compared to the wild-type strain (Figure 2B), indicating that the mutants gained additional lactic acid tolerance during adaptive evolution.

### 3.2. Whole-Genome Sequence Analysis of K. pneumoniae (ATCC25955)

To determine the mutation location of these evolved strains, we sequenced the whole genome of *K. pneumoniae* (ATCC25955) wild-type for the first time using a Pacbio sequel II and DNBSEQ 2000 platform. To improve the accuracy of the genome sequences, GATK (v1.6-13) was used to make single-base corrections. The complete genome of *K. pneumoniae* (ATCC25955) consists of a single circular chromosome of 5,107,238 bp with an average GC content of 57.59% (Figure 3A) and contains a circular plasmid with a length of 232,387 bp, with a GC content of 52.70% (Figure 3B). A total of 5076 CDSs, 86 tRNAs, 25 rRNAs, and 53 sRNAs were detected in the genome, and the basic characteristics of the genome are shown in Appendix A. KEGG (Kyoto Encyclopedia of Genes and Genomes), COG (Clusters of Orthologous Groups), and GO (Gene Ontology) are used for general function annotation. The KEGG metabolic pathway includes human cellular processes, environmental information processing, genetics, human diseases, metabolism, and organismal systems (Appendix A). The largest number of identified genes was classified into metabolism pathways. Among the genes annotated with COG function, the most related genes are involved in metabolism (Appendix A). The molecular functions, cellular components, and biological processes of these genes were elucidated by GO terms, and the genes involved in the biological process were most abundant (Appendix A).

### 3.3. Genome Resequencing Analysis of Evolved Strains

To understand how the mutants had evolved to acquire enhanced lactic acid tolerance, the genetic alterations in the genomes of the eight mutants were analyzed using a DNBSEQ platform. Comparative analysis of incomplete genome sequences of the eight mutants with that of the wild type identified a total of four single nucleotide polymorphism (SNP) mutations (Table 2), including two intergenic SNP mutations and two SNP mutations in the coding region. All developed strains contain a single SNP at location 464,716 in the genomic region upstream of 23S rDNA. The strains C8 and D3 exhibit an identical SNP mutation in the *rcsA*-*yedD* intergenic region. Furthermore, SNP mutations in the coding regions of the *glaU* and *ydhS* genes have been detected in the A1 evolved strain, as well as the A2, A3, A4, and A8 evolved strains, respectively (Table 2). These genetic insights provide valuable clues for understanding the adaptive mechanisms that contribute to lactic acid tolerance in *K. pneumoniae*.

### 3.4. Intergenic SNP Mutations in Evolved Strains Affect Gene Transcription

To explore the effect of intergenic SNP mutations on the lactic acid tolerance of *K. pneumoniae*, we first examined the expression of target genes in evolved strains and wild-type strains with or without lactic acid challenge. The qRT-PCR analysis revealed that the transcription levels of 23s rDNA in most evolved strains exhibited a slight decrease compared with the wild-type strain under the lactic acid challenge (Figure 4A). Notably, 23S rRNA plays a critical role in ensuring proper growth and multidrug resistance. Liiv et al. [28] suggested that mutations of 23S rRNA severely limit translation and influence growth to variable degrees. Martini et al. [29] have demonstrated that the deletion of a critical RNase J for 23S rRNA maturation leads to multidrug resistance in *Mycobacterium tuberculosis*. In addition, an atypical mutation in *Neisseria gonorrhoeae* 23S rRNA is associated with high-level azithromycin resistance [30]. However, more research is needed to determine whether it contributes to lactic acid resistance.

In addition, the transcription level of the *yedD* gene in evolved strains C8 and D3 did not present significant differences compared with the starting strain with or without lactic acid challenge, whereas the transcription level of the *rcsA* gene was much lower (Figure 4B). RcsA, a coregulator of the RCS two-component system, is associated with capsular polysaccharide and biofilm formation and mediates bacterial survival under several stressful conditions [31]. To delve deeper into the impact of the SNP mutation in the *rcsA*-*yedD* intergenic region on gene transcription, we constructed a mutant strain with the SNP at position 1780734. Our qRT-PCR analysis revealed that the transcriptional level of *rcsA* decreased in this SNP mutant strain, regardless of lactic acid stress, when compared to the wild-type (Figure 4C).

### 3.5. galU Is Associated with Increased Lactic Acid Tolerance

To confirm the relation between the candidate genes and the enhanced lactic acid tolerance of the mutants, a survival assay after the lactic acid challenge was carried out. Given that the evolved strains C8 and D3 had considerably altered *rcsA* transcription levels in comparison to the wild-type strain, we investigated the potential involvement of *rcsA* in lactic acid tolerance. Surprisingly, the survival of the ∆*rcsA* strain did not significantly differ from that of the wild-type strain after the lactic acid challenge (Figure 5A). In summary, although whole-genome sequencing revealed an SNP in the *rcsA*-*yedD* intergenic region, this mutant impact on *rcsA* transcription—whether in the presence or absence of lactic acid stress—may not be exclusively related to lactic acid stress responses.

Moving forward, we explored the effect of SNP mutations within coding regions of the *galU* and *ydhS* genes in the evolved strains. Intriguingly, only the deletion of *galU* dramatically lowered *K. pneumoniae*’s lactic acid tolerance (Figure 5A). To confirm the function of *galU*, we performed complementation experiments (Figure 5B), conclusively demonstrating that the *galU* gene plays a pivotal role in conferring lactic acid resistance of *K. pneumoniae*. In addition, the qRT-PCR analysis revealed that the transcription levels of *galU* in evolved strain A1 exhibited a significant increase compared with the wild-type strain regardless of being exposed to the lactic acid challenge (Appendix A). Surprisingly, although *galU* is essential for bacterial lactic acid tolerance, its overexpression could not contribute to bacterial survival under lactic acid stress and might even create a growth burden due to protein overproduction.

GalU (UTP—glucose-1-phosphate uridylyltransferase) catalyzes the production of UDP-glucose from a UTP molecule and glucose-1-phosphate [32]. GalU and UDP-glucose are required for the manufacture of bacterial glycopolymers, including capsular polysaccharide, teichoic acid, and lipopolysaccharide (LPS), which constitute the fundamental components that determine the characteristics of the bacterial cell surface [33]. Studies have reported that GalU has a crucial role in biofilm formation and cell wall biosynthesis [34,35,36]. Indeed, biofilms can confer cell adaptability to environmental change and higher resistance to adverse conditions, which is a self-protective mechanism for growth [37,38]. Furthermore, the cell wall functions as the main sensory interface that is necessary for interactions with the outside world and survival, and it is involved in resistance mechanisms against multiple environmental stresses [39,40]. GalU has been revealed that it plays a role in many bacteria’s environmental stress responses. For example, Serror et al. [41] discovered that inactivating *galU* reduces cellular resistance to a variety of stressors, including antibiotics and H_2_O_2_. Kurushima et al. [36] demonstrated that *galU* was essential for enterococcal polysaccharide antigen (EPA) biosynthesis in *E. faecalis*. Studies have suggested that EPA deficit causes diminished cell surface integrity, which increases sensitivity to antimicrobial agents or several environmental stresses [42,43,44]. Our study sheds light on the novel implication of the *galU* gene in bacterial lactic acid tolerance. The susceptibility of *K. pneumoniae* to lactic acid may indeed be linked to impaired envelope and cell wall function mediated by GalU. However, further research is warranted to unravel the precise mechanisms underlying this susceptibility and explore strategies for enhancing lactic acid tolerance in robust lactic acid-producing microorganisms.

### 3.6. High d-Lactic Acid Production in Evolved Strains

In our previous work, engineered *K. pneumonia* was constructed by deleting acetate kinase (*ackA*), alcohol dehydrogenase (*adhE*), and acetolactic acid synthase (*budB*). The resultant strain Q2702 produced an extraordinary amount of d-lactic acid from glucose with pH adjustment, which was much higher than that of the wild-type strain, and the optical purity was almost 100% [6]. Building upon this success, we selected A4 and B1 with higher lactic acid resistance along with *galU* mutation-related strains (designated as A1) as starting strains, then all of them knocked out *budB*, *ackA*, and *adhE* to create three engineered strains: Q5221, Q5224, and Q5227. These strains demonstrated increased tolerance to lactic acid-containing media (Figure 6A). Further testing is needed to evaluate their d-lactic acid production capabilities and overall fitness compared to the starting strains. Indeed, we intend to obtain engineering *K. pneumoniae* strains that could tolerate both low pH and lactate, resulting in increased lactic acid production without pH adjustment. For example, in our previous study, overexpressing the *fabA* gene could enhance *E. coli* tolerance to organic acid environments, leading to higher 3-hydroxypropionate production without pH adjustment [45]. As a result, the need for additional acid and base titrant use will be avoided, thus reducing the overall cost of production. Unfortunately, without pH adjustment, all of our evolved engineered strains (Q5221, Q5224, and Q5227) and Q2702 produced little d-lactic acid, and these strains showed no significant difference in cell growth. However, these engineered strains exhibited equivalent or superior overall growth ability and higher d-lactic acid production compared to the control strain Q2702 with pH adjustment by NH_3_·H_2_O (Figure 6B,C). Notably, strain Q5224 stood out, achieving exceptional growth activity and the highest d-lactic acid production of 19.56 g/L in shake flask fermentation (Figure 6B,C). In addition, the wild-type strain was basically unable to produce d-lactic acid (Appendix A), which is consistent with our previous study [6]. Given that the inhibition caused by lactic acid cannot be merely attributed to proton release within the cytoplasm, even though it causes a significant decrease in intracellular pH. The accumulation of the acid anion also has significant effects on cell physiology. Indeed, all engineered strains (Q5221, Q5224, and Q5227) with significantly increased lactic acid tolerance exhibited no difference under hydrogen chloride challenge at the same acid pH (Appendix A). As a consequence, we propose that the evolutionary strains may have developed tolerance to lactate ions rather than low pH. This adaptation enables improved growth and enhanced production in neutral fermentation media with high lactic acid concentrations.

In addition, diverse carbon sources have been employed in d-lactate fermentation, including glucose, glycerol, sweet potato, sugar beet pulp, and corncob slurry. In recent years, studies about engineered *Klebsiella pneumonia*, *Saccharomyces cerevisiae*, *Escherichia coli*, *Lactobacillus coryniformis*, *Pichia kudriavzevii* and some other microorganisms for producing d-lactate have been reported. We have compared the d-lactate fermentation capabilities of different strains using different carbon sources in flask fermentation (Table 3). To our knowledge, the production of 19.56 g/L in this study represents the highest d-lactic acid production in flask shake by *K. pneumoniae* to date. Notably, d-lactic acid fermentation is mainly cultured in a complex medium containing broth or yeast at a relatively high cost. As a result, the high productivity and optical purity of d-lactic in a low-cost, minimal medium in this study indicated that the engineered *K. pneumoniae* Q5224 is an excellent producer of d-lactate.

## 4. Conclusions

In our study, we conducted ALE experiments intending to generate new lactic acid-tolerant strains capable of efficiently fermenting d-lactic acid in *K. pneumoniae*. The results revealed remarkable improvements in lactic acid tolerance in our endpoint strains. Notably, their improved resistance to lactic acid was primarily due to their acquisition of lactate tolerance rather than low pH, which resulted in enhanced d-lactic acid production. Additionally, we identified a novel gene related to lactic acid tolerance, *galU*. In conclusion, adaptive laboratory evolution has been demonstrated as an effective strategy for enhancing d-lactic acid production. The uncovered tolerance response to high lactic acid concentration, resulting from genetic mutations in the evolved strains, provides valuable guidance for the future engineering of commercial d-lactic acid production.

## Figures and Tables

**Figure 1 microorganisms-12-01167-f001:**
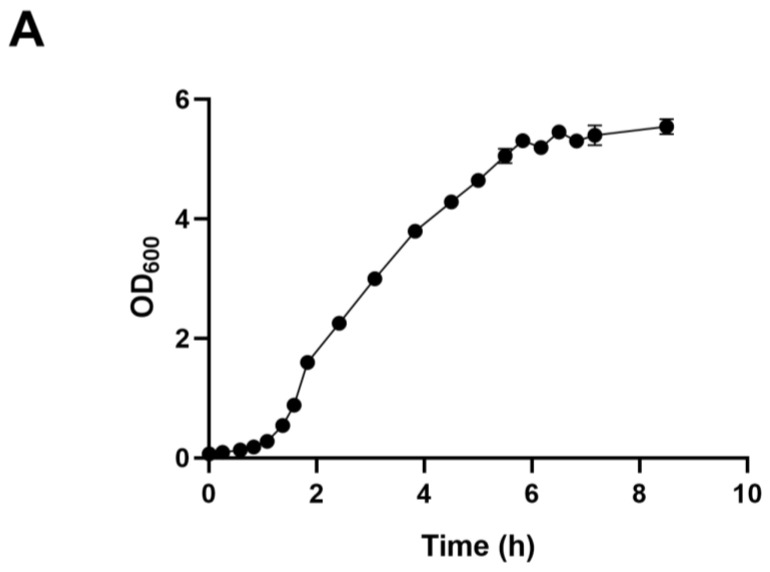
Lactic acid tolerance of *K. pneumoniae* (ATCC25955) wild-type. (**A**) The cell growth of *K. pneumoniae* (ATCC25955) wild-type in lactic acid fermentation medium. (**B**) The cell growth of *K. pneumoniae* (ATCC25955) wild-type in a lactic acid fermentation medium supplied with indicated concentrations of lactic acid. (**C**) The cell survival of *K. pneumoniae* (ATCC25955) wild-type in lactic acid fermentation medium supplied with indicated concentrations of lactic acid. Error bars indicate the standard deviations of three independent experiments.

**Figure 2 microorganisms-12-01167-f002:**
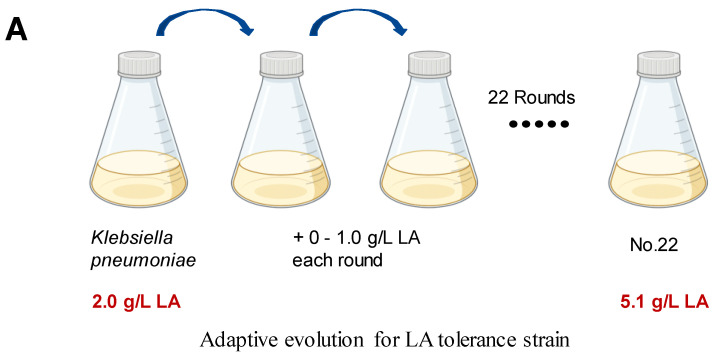
Adaptive laboratory evolution enhances lactic acid tolerance of *K. pneumoniae*. (**A**) ALE of strain *K. pneumoniae* (ATCC25955) in lactic acid fermentation medium supplemented with increasing concentrations of LA from 2.0 g/L to 5.1 g/L. (**B**) lactic acid susceptibility assay for the *K. pneumoniae* (ATCC25955) wild-type strain and eight evolved strains after lactic acid challenge at 0 or 3.0 g/L for 4 h. The corresponding survival of *K. pneumoniae* (ATCC25955) wild-type strain was used as the control. Statistical analysis was performed using a two-tailed Student’s *t*-test (*** *p* < 0.001, **** *p* < 0.0001).

**Figure 3 microorganisms-12-01167-f003:**
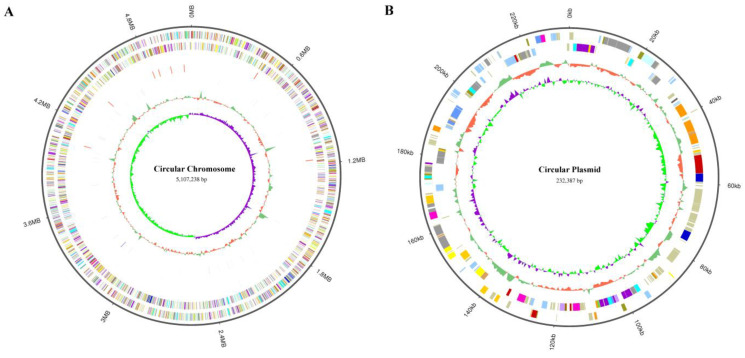
Visualization of *K. pneumoniae* (ATCC25955) whole-genome Sequence by using Circos (v 0.69-8). (**A**) The genome of circular chromosome (5,107,238 bp) in *K. pneumoniae* (ATCC25955). It shows several layers, starting from the outer layer representing the genome size, the forward strand gene, the reverse strand gene, the forward strand ncRNA, the reverse strand ncRNA, the repeat, GC, and GC-SKEW. (**B**) The genome of circular plasmid (232,387 bp) in *K. pneumoniae* (ATCC25955). It shows several layers, starting from the outer layer representing the genome size, the forward strand gene, the reverse strand gene, GC, and GC-SKEW.

**Figure 4 microorganisms-12-01167-f004:**
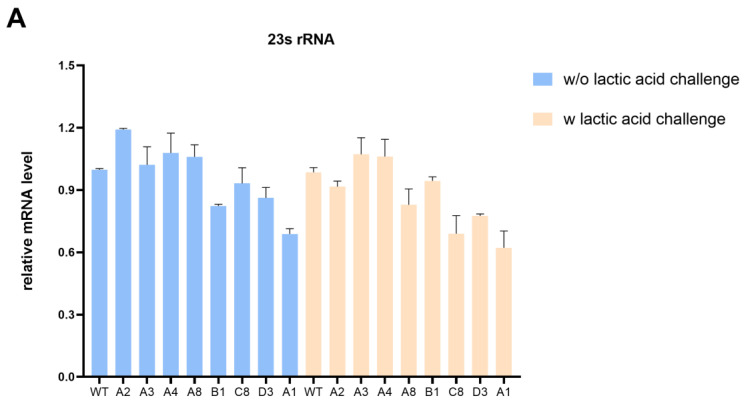
Intergenic SNP mutations in evolved strains affect gene transcription. (**A**) The impact of mutations in evolved strains on transcription of 23s rRNA with or without lactic acid challenge. (**B**) The impact of mutations in C8 and D3 on the transcription of *rcsA* and *yedD* with or without lactic acid challenge. (**C**) The impact of the SNP mutation in the *rcsA*-*yedD* intergenic region on transcription of *rcsA* with or without lactic acid challenge.

**Figure 5 microorganisms-12-01167-f005:**
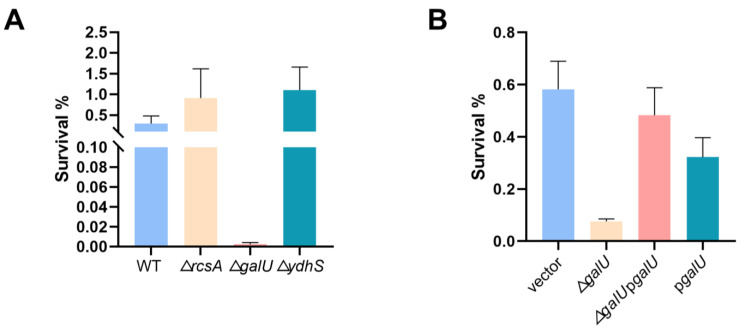
*galU* is implicated in the lactic acid tolerance of *K. pneumoniae*. (**A**) Lactic acid susceptibility of wild-type and ∆*rcsA*, ∆*ydhS*, ∆*galU* strains after stimulation with or without 3 g/L lactic acid for 4 h (*n* = 3 biologically independent samples). (**B**) Lactic acid susceptibility assay of *K. pneumoniae* (ATCC25955) carrying empty vector or p*galU* and ∆*galU* mutant carrying empty vector or p*galU* strains after stimulation with or without 3 g/L lactic acid for 4 h. Statistical analysis was performed using a two-tailed Student’s *t*-test.

**Figure 6 microorganisms-12-01167-f006:**
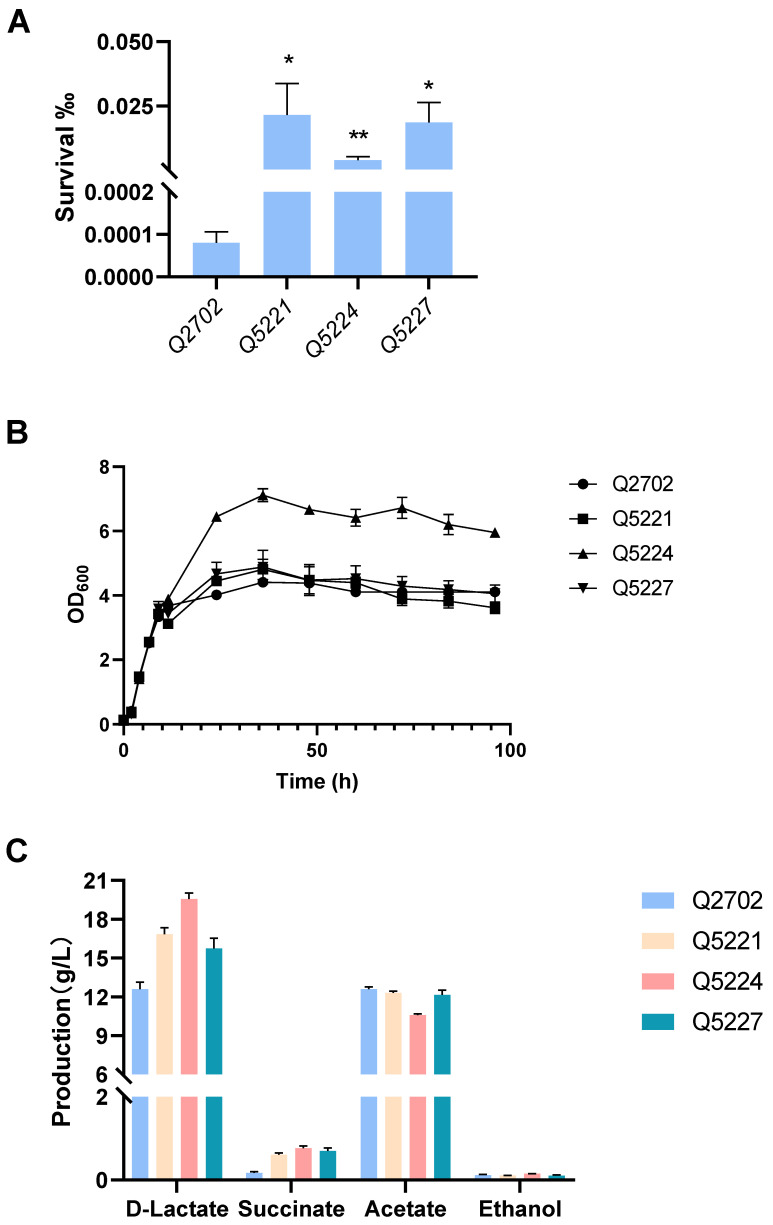
Lactic acid susceptibility assay, cell growth, and d-lactic acid production in shake flask fermentation. (**A**) Lactic acid susceptibility assay of the control strain Q2702 and the engineered strains in the fermentation media after stimulation with or without 3 g/L lactic acid for 2 h (* *p* < 0.1, ** *p* < 0.01). (**B**) Growth curves of the control strain Q2702 and the engineered strains acquired in this study with glucose as the carbon source. (**C**) d-lactic acid and by-products produced in shake flask fermentation.

**Table 2 microorganisms-12-01167-t002:** Mutations of evolved strains identified by genome sequencing.

Gene ID	Gene Name	Protein Description	Position	Nucleotide Alteration	Amino Acid Alteration	Strain
GL000448	-	hypothetical protein	464716	C→T	Intergenic SNP mutation	A1, A2, A3, A4, A8, B1, C8, D3
GL000449	-	hypothetical protein				
GL001697	*rcsA*	transcriptional regulator RcsA	1780734	A→G	Intergenic SNP mutation	C8, D3
GL001698	*yedD*	lipoprotein				
GL001910	*galU*	UTP-glucose-1-phosphate uridylyltransferase GalU	2002430	C→A	A→E	A1
GL002357	*ydhS*	FAD-NAD(P)-binding protein	2455010	C→A	P→Q	A2, A3, A4, A8, B1

**Table 3 microorganisms-12-01167-t003:** Comparison of d-lactate production by different strains using different carbon sources in flask fermentation.

Organism	Minimal Medium	Carbon Source	d-Lactic Acid Production (g/L)	Optical Purity (%)	References
*Klebsiella pneumoniae*	Yes	Glucose	19.56	~100	This study
*Klebsiella pneumoniae*	No	Glucose	14.08	~100	[46]
*Klebsiella pneumoniae*	No	Glycerol	8.33	~100	[11]
*Saccharomyces cerevisiae*	Yes	Glucose	17.09	~100	[47]
*Saccharomyces cerevisiae*	No	Glucose	11.14	~100	[48]
*Escherichia coli*	No	Glucose	16.20	NA	[49]
*Lactobacillus coryniformis*	No	Sugar beet pulp	19.30	99.5	[50]
*Lactobacillus coryniformis*	No	Glucose	19.70	~100	[51]
*Lactobacillus saerimneri*	No	Sucrose	~15.00	NA	[52]
*Leuconostoc*	No	Sugar beet pulp	14.00	93.9	[50]
*Pichia kudriavzevii*	No	Glucose	62.00	NA	[53]
*Pediococcus acidilactici*	No	Corncob slurry	61.90	NA	[54]

## Data Availability

The original contributions presented in the study are included in the article/Appendix A, further inquiries can be directed to the corresponding authors.

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
