# Peer review of "Adaptive Evolution for the Efficient Production of High-Quality d-Lactic Acid Using Engineered Klebsiella pneumoniae"

_microorganisms, 2024, doi:10.3390/microorganisms12061167_

Round 1
Reviewer 1 Report
Comments and Suggestions for Authors
The manuscript by Bo Jiang and co-workers describes the development of lactate hypertolerant strains of K. pneumoniae by adaptive laboratory evolution, identification of gene mutations present in evolved strains and the ability of evolved strains to produce higher amounts of lactic acid.
The study is interesting and, in general, well organized. The d-lactate titer produced by the evolved strain is still low for commercial application. Il will be interesting to know which is the production of the WT strain to know how much it was improved. In addition, a number of aspects in the experimental procedures need to be clarified or implemented as well as in text text/figures as detailed below.
Major points
1. Lines 71-76. Recent studies reporting the use of ALE for improving microbial tolerance to lactate also include: Mazzoli et al., 2022. N. Biotechnol. 67, 12–22. https://doi.org/10.1016/j.nbt.2021.12.003; Svetlitchnyi et al., 2022. Biotechnol. Biofuels Bioprod. 15, 44. https://doi.org/10.1186/s13068-022-02137-7)
2. Table 2 is cited before table 1
3. Table 2. I think more information (which research center ?) should be indicated for Dr. Roy Curtiss III
4. Table S1: the table legend should describe why capital letters are sometimes used for primer nucleotides
5. Section 2.3. For adaptive evolution, was lactic acid used or lactate salt ?
6. Lines 144-145. What is PG tolerance ? What is BGI ?
7. Figure 2B and caption. The % of survival seems very low: is that correct ? The strategy used to test lactic acid susceptibility described here does not seem to correspond to what is described in the materials and methods (Section 2.2). Please, clarify. Was lactic acid used for tolerance test or some lactate salt ? If lactic acid was used did Author take into account the effect of its supplementation on pH ? “*P < 0.05, **P < 0.01” is useless since this is not present in the figure
8. Line 234. Why the level of transcription of 23s rDNA was analyzed only in mutants C8 and D3 since the mutation in the upstream region is present in all the mutant strains ? I think this information is very important to confirm that the mutation actually lower gene transcription. In addition, why the level of transcription was measured only under lactic acid stress ? It is interesting to know if this change also causes alteration of 23rDNA transcription also in control conditions.
9. Fig. 4. Caption lacks description for Fig. 4B
10. Lines 260-268. These lines seem to better fit with section 3.4 than 3.5. Probably the title of section 3.4. should be changed since it only describes the effect of mutations in intergenic regions upstream of 23s rDNA and between rcsA-yedD.
11. Lines 269-295. The effect of inactivation of galU e ydhS on tolerance to lactic acid was investigated. These data are interesting, however, in lactic acid tolerant mutants of K. pneumoniae gene mutations generated amino acid replacement and increased tolerance. Did author try to re-introduce these mutations ? What these mutations are supposed to cause on the activity of these proteins ? Did they occur in specific domains of the proteins ?
12. Fig. 5B. The figure seems to lack a bar (WT strain + pgalU)
13. Lines 308-309. The sentence is not clear. I understand from table than only strain A1 show the galU mutation, A4 is the strain with the highest lactate tolerance while it’s not clear why strain B1 was selected.
14. Lines 310-311. Knock-out of the budB, ackA, and adhE was performed in all the 3 engineered strains ?
15. Lines 318-328. Also other strains which were evolved towards higher lactate tolerance did not show higher acidic pH tolerance. (Please see Mazzoli et al., 2022. N. Biotechnol. 67, 12–22. https://doi.org/10.1016/j.nbt.2021.12.003; Svetlitchnyi et al., 2022. Biotechnol. Biofuels Bioprod. 15, 44. https://doi.org/10.1186/s13068-022-02137-7). Organic acid toxicity is actually due both to effects on pH and effects related to the anion (see Mazzoli, R., 2021. Fermentation 7, 248. https://doi.org/10.3390/fermentation7040248)
16. Fig. 6C. How much lactate can produce the wild type strain in the same conditions (regulated pH) ?
Author Response
Response to Reviewer 1:
We thank you for your constructive comments and appreciation for the importance of our study. We have carefully revised the manuscript and uploaded a compared copy of the manuscript (without figures) as a "Marked-Up Manuscript" file (changes in the manuscript are marked in red), and here are point-by-point responses:
Major points
- Lines 71-76. Recent studies reporting the use of ALE for improving microbial tolerance to lactate also include: Mazzoli et al., 2022. N. Biotechnol. 67, 12–22. https://doi.org/10.1016/j.nbt.2021.12.003; Svetlitchnyi et al., 2022. Biotechnol. Biofuels Bioprod. 15, 44. https://doi.org/10.1186/s13068-022-02137-7)
Reply: Thank you for your precious suggestion. We have scrutinized these papers and found these studies are quite important for improving microbial tolerance to lactate. We have added this part (Marked-Up Manuscript, lines 77-81), and now it reads:
“Mazzoli et al[16] increased the lactic acid tolerance of C. thermocellum strains LL345 (parent strain) and LL1111 (ΔadhE, lactic acid overproducing strain) through ALE. Svetlitchnyi et al[17] obtained evolved strains originating from Caldicellulosiruptor sp. DIB 104C by ALE, which could produce unusually large amounts of l-lactic acid from micro-crystalline cellulose in fermenters.”
- Table 2 is cited before table 1
Reply: Thank you for your precious suggestion. We have adjusted the order of these tables and revised correlative sentence in the revised manuscript (Marked-Up Manuscript, line 107, line 253, and line 258).
- Table 2. I think more information (which research center ?) should be indicated for Dr. Roy Curtiss III
Reply: We sincerely apologize for providing unclear information on the source of E. coli χ7213. We have reinterpreted its source information and attached the corresponding paper in Table 1 (the previous Table 2).
- Table S1: the table legend should describe why capital letters are sometimes used for primer nucleotides
Reply: Sorry for the mixed uppercase and lowercase letters of primer nucleotides in Table S1. We have modified Table S1.
- Section 2.3. For adaptive evolution, was lactic acid used or lactate salt ?
Reply: We sincerely apologize for the unclear description of adaptive evolution. we conducted an adaptation experiment by gradually increasing lactic acid rather than lactate salt.
- Lines 144-145. What is PG tolerance ? What is BGI ?
Reply: We sincerely apologize for the Irregular writing. We have revised “PG” to “lactic acid” and added the full name of BGI in the revised manuscript (Marked-Up Manuscript, lines 163-164), and now it reads:
“Eight colonies with the highest lactic acid tolerance were designated as A1, A2, A3, A4, A8, B1, C8, and D3, and their genomes were resequenced by Beijing Genomics institution (BGI).”
- Figure 2B and caption. The % of survival seems very low: is that correct ? The strategy used to test lactic acid susceptibility described here does not seem to correspond to what is described in the materials and methods (Section 2.2). Please, clarify. Was lactic acid used for tolerance test or some lactate salt ? If lactic acid was used did Author take into account the effect of its supplementation on pH ? “*P < 0.05, **P < 0.01” is useless since this is not present in the figure
Reply: We apologize for the unclear explanation.
Firstly, in this study, survival is CFU ratio that represents the lactic acid tolerance of exponentially growing bacterial. Survival (%) = (CFU with lactic acid challenge/CFU without lactic acid challenge) × 100%. We added this formula (Marked-Up Manuscript, line 154) and rewrote the caption of Figure 2B (Marked-Up Manuscript, lines 217-218). After 4 hours, “CFU without lactic acid challenge” grew dramatically, resulting in a high denominator and a very low survival (%).
Secondly, we used lactic acid for tolerance test. Indeed, we intend to enhance K. pneumoniae tolerance to both low pH and lactate, leading to higher lactic acid production without pH adjustment. For example, in our previous study, overexpressing fabA gene could enhance E. coli tolerance to organic acid environments, leading to similar 3-hydroxypropionate production and cell growth with and without pH adjustment (Xu Y, et al. Nat Commun. 2020 Mar 20;11(1):1496. doi: 10.1038/s41467-020-15350-5). Unfortunately, without any pH adjustment, all of our evolved engineered strains (Q5221, Q5224, and Q5227), and the control strain Q2702 produced little d-lactic acid. Only with pH adjustment, these engineered strains exhibited higher d-lactic acid production compared to the control strain (Figure 6C). In addition, all engineered strains (Q5221, Q5224, and Q5227) demonstrated increased tolerance to lactic acid-containing media (Figure 6A). However, all engineered strains (Q5221, Q5224, and Q5227) with significantly increased lactic acid tolerance exhibited no difference under hydrogen chloride (HCl) challenge at the same acid pH (Figure S6). As a consequence, we propose that the evolutionary strains may have developed tolerance to lactate ions rather than low pH. This adaptation enables improved growth and en-hanced production in neutral fermentation media with high lactic acid concentrations. We rewrote this part in the revised manuscript (Marked-Up Manuscript, lines 347-373).
Thirdly, the corresponding survival of K. pneumoniae (ATCC25955) wild-type strain was used as the control. Statistical analysis was performed using a two-tailed Student’s t-test (*P < 0.05, **P < 0.01, ***P < 0.001, ****P < 0.0001) (Marked-Up Manuscript, lines 218-219).
- Line 234. Why the level of transcription of 23s rDNA was analyzed only in mutants C8 and D3 since the mutation in the upstream region is present in all the mutant strains ? I think this information is very important to confirm that the mutation actually lower gene transcription. In addition, why the level of transcription was measured only under lactic acid stress ? It is interesting to know if this change also causes alteration of 23rDNA transcription also in control conditions.
Reply: Thank you a lot for your precious suggestion. We fully agree that the level of transcription of 23s rDNA should be analyzed in all the mutants and we sincerely apologize for the lacking. We have supplemented the relevant analysis and updated Figure 4 in the revised manuscript. The qRT-PCR analysis revealed that the transcription levels of 23s rDNA in most evolved strains exhibited a slight decrease compared with the wild-type strains under the lactic acid challenge (Figure 4A). However, more research is needed to determine whether it contributes to lactic acid resistance. We have rewritten this paragraph in the revised manuscript (Marked-Up Manuscript, lines 267-273).
- Fig. 4. Caption lacks description for Fig. 4B
Reply: We sincerely apologize for the caption lacks description for Fig. 4B. We have written this part in the revised manuscript (Marked-Up Manuscript, lines 296-301), and now it reads:
“Figure 4. Intergenic SNP mutations in evolved strains affect gene transcription. (A) The impact of mutations in evolved strains on transcription of 23s rRNA with or without lactic acid challenge. (B) The impact of mutations in C8 and D3 on the transcription of rcsA and yedD with or without lactic acid challenge. (C) The impact of the SNP mutation in the rcsA-yedD intergenic region on transcription of rcsA with or without lactic acid challenge.”
- Lines 260-268. These lines seem to better fit with section 3.4 than 3.5. Probably the title of section 3.4. should be changed since it only describes the effect of mutations in intergenic regions upstream of 23s rDNA and between rcsA-yedD.
Reply: Thank you for your precious suggestion. We changed the title of section 3.4 to “Intergenic SNP mutations in evolved strains affect gene transcription”.
In addition, we fully agree that lines 260-268 (Marked-Up Manuscript, lines 289-301) are better fit with “section 3.4”. “Section 3.4” intends to explore the effect of intergenic SNP mutations on the lactic acid tolerance of K. pneumoniae, we first examined the expression of target genes in evolved strains and wild-type strains. Due to the transcription level of rcsA significantly reduced in the evolved strains, to delve deeper into the impact of the SNP mutation in the rcsA-yedD intergenic region on gene transcription, we constructed a mutant strain with the SNP at position 1780734 and examined the expression of rcsA with or without lactic acid challenge.
11. Lines 269-295. The effect of inactivation of galU e ydhS on tolerance to lactic acid was investigated. These data are interesting, however, in lactic acid tolerant mutants of K. pneumoniae gene mutations generated amino acid replacement and increased tolerance. Did author try to re-introduce these mutations ? What these mutations are supposed to cause on the activity of these proteins ? Did they occur in specific domains of the proteins ?
Reply: Thank you very much for your precious suggestion. In our study, the SNP mutation in the galU gene generates amino acid replacement and increases the lactic acid tolerance of K. pneumoniae. We wholeheartedly concur that it is critical to investigate how this mutation affects protein activity and whether replacing amino acids at other locations can likewise have an impact on lactic acid tolerance. Future research will carry out further exploration of this section.
12. Fig. 5B. The figure seems to lack a bar (WT strain + pgalU)
Reply: Thank you for your precious suggestion. We completely agree that Figure 5B should show this bar. We added the bar of overexpression galU in Figure 5B. Surprisingly, although galU is essential for bacterial lactic acid tolerance, its overexpression could not contribute to bacterial survival under lactic acid stress and might even create a growth burden due to protein overproduction. We added this sentence in the revised manuscript (Marked-Up Manuscript, lines 319-322).
13. Lines 308-309. The sentence is not clear. I understand from table than only strain A1 show the galU mutation, A4 is the strain with the highest lactate tolerance while it’s not clear why strain B1 was selected.
Reply: We sincerely apologize for the unclear description of how these candidates were chosen. In this study, we selected A4 and B1 with higher lactic acid resistance along with galU mutation-related strains (designated as A1) as starting strains. We have rewritten this sentence in the revised manuscript (Marked-Up Manuscript, lines 358-359).
14. Lines 310-311. Knock-out of the budB, ackA, and adhE was performed in all the 3 engineered strains?
Reply: We sincerely apologize for the unclear explanation. In this study, we selected A1, A4 and B1 as starting strains, then all of them knocked out budB, ackA, and adhE to create three engineered strains: Q5221, Q5224, and Q5227. We have rewritten this sentence in the revised manuscript (Marked-Up Manuscript, line 360).
15. Lines 318-328. Also other strains which were evolved towards higher lactate tolerance did not show higher acidic pH tolerance. (Please see Mazzoli et al., 2022. N. Biotechnol. 67, 12–22. https://doi.org/10.1016/j.nbt.2021.12.003; Svetlitchnyi et al., 2022. Biotechnol. Biofuels Bioprod. 15, 44. https://doi.org/10.1186/s13068-022-02137-7). Organic acid toxicity is actually due both to effects on pH and effects related to the anion (see Mazzoli, R., 2021. Fermentation 7, 248. https://doi.org/10.3390/fermentation7040248)
Reply: Thank you for your precious information. We have scrutinized these papers.
Firstly, Mazzoli et al's research are quite important for improving C. thermocellum tolerance to lactate. We cited this paper in the revised manuscript (Marked-Up Manuscript, lines 70-72). In their study, sodium lactate was used instead of lactic acid to avoid medium acidification.
Secondly, we fully agree that “organic acid toxicity is actually due both to effects on pH and effects related to the anion”. We cited this paper in the revised manuscript (Marked-Up Manuscript, lines 52-53).
In our study, we used lactic acid for ALE rather than lactate. We intend to enhance K. pneumoniae tolerance to both low pH and lactate, leading to higher lactic acid production without pH adjustment. Unfortunately, without any pH adjustment, the engineered strains produced no more D-lactic acid than the control strains. However, these engineered strains exhibited higher D-lactic acid production with pH adjustment (Figure 6C). Indeed, all engineered strains (Q5221, Q5224, and Q5227) with significantly increased lactic acid tolerance exhibited no difference under hydrogen chloride challenge at the same acid pH (Figure S6). As a consequence, we propose that the evolutionary strains may have developed tolerance to lactate ions rather than low pH. This adaptation enables improved growth and enhanced production in neutral fermentation media with high lactic acid concentrations. We rewrote this part in the revised manuscript (Marked-Up Manuscript, lines 363-389).
16. Fig. 6C. How much lactate can produce the wild type strain in the same conditions (regulated pH)?
Reply: Sorry for this lacking information. We fully agree that the production of wild type strains should be analyzed in the same conditions (regulated pH). We have supplemented this data and added Figure S4 in the revised manuscript. The wild-type strain was basically unable to produce D-lactic acid, which is consistent with our previous study (Feng X et al. Microb Cell Fact. 2017 Nov 21;16(1):209. doi:
10.1186/s12934-017-0822-6.). We have added this sentence in the revised manuscript (Marked-Up Manuscript, lines 378-379).

Reviewer 2 Report
Comments and Suggestions for Authors
Major comments:
1. Figure 1C, why not measure the CFU at 0 h?
2. Since GalU is associated with increased lactic acid tolerance, why not analysis the transcription levels of galU in Results 3.4?
3. I recommend overexpressing the mutate galU using a plasmid in results 3.5, whether it can further enhance the lactic acid tolerance?
4. The production of D-lactic acid is very low to my knowledge, why not measure and compare the D-lactic acid production of the Fed batch fermentations? Because in other articles published by the same authors, the strain can accumulate 125.1 and 142 g/L d-lactate.
Minor comments:
1. What is the mean of high-quality D-lactic acid in the title? I am not able to find the purity of D-lactate in the whole results.
2. Line 16, “productivity” is not accurate, please change to production or yield.
3. Line 39, delete “can be generated”
4. Line 42, it seems the fermentation medium is very complicated and contains yeast extract, why the authors think it is a minimal medium?
5. Line 98, 37 ℃
6. Line 114-115, suicide plasmids
7. “OD600” in the text and figure should be unified.
Author Response
Response to Reviewer 2:
We thank you for your constructive comments for our study. We have carefully revised the manuscript and uploaded a compared copy of the manuscript (without figures) as a "Marked-Up Manuscript" file (changes in the manuscript are marked in red), and here are point-by-point responses:
Major comments:
- Figure 1C, why not measure the CFU at 0 h?
Reply: Sorry for the lack of “the CFU at 0 h” in Figure 1C. We have supplemented this data and readjusted Figure 1C in the revised manuscript.
- Since GalU is associated with increased lactic acid tolerance, why not analysis the transcription levels of galUin Results 3.4?
Reply: We sincerely apologize for the misleading information in “Results 3.4”. Indeed, “Results 3.4” intends to explore the effect of intergenic SNP mutations on the lactic acid tolerance of K. pneumonia. We first examined the expression of target genes in evolved strains and wild-type strains. Thank you very much for your precious suggestion. We fully agree that we should analyze the transcription levels of galU. We have supplemented the relevant analysis and added Figure S5 in the revised manuscript. As we expected, the qRT-PCR analysis revealed that the transcription levels of galU in evolved strain A1 exhibited a significant increase compared with the wild-type strain regardless of being exposed to the lactic acid challenge, demonstrating its role in conferring lactic acid resistance of K. pneumoniae (Marked-Up Manuscript, lines 317-319).
- 3. I recommend overexpressing the mutate galU using a plasmid in results 3.5, whether it can further enhance the lactic acid tolerance?
Reply: Thank you for your precious suggestion. We completely agree that “Results 3.5” should show an analysis of whether overexpressing galU can further enhance lactic acid tolerance. We added the bar of pgalU in Figure 5B. Surprisingly, although galU is essential for bacterial lactic acid tolerance, its overexpression could not contribute to bacterial survival under lactic acid stress and might even create a growth burden due to protein overproduction. We added this sentence in the revised manuscript (Marked-Up Manuscript, lines 319-322).
- The production of D-lactic acid is very low to my knowledge, why not measure and compare the D-lactic acid production of the Fed batch fermentations? Because in other articles published by the same authors, the strain can accumulate 125.1 and 142 g/L d-lactate.
Reply: Thank you a lot for your precious suggestion. We fully agree that fed-batch fermentations would further increase the d-lactic acid production. In our previous studies, we knocked out by-product synthesis genes or overexpressed d-lactic acid dehydrogenase that can produce 125.1 and 142 g/L d-lactic acid from glucose and glycerol, respectively (Feng X et al. Microb Cell Fact. 2017 Nov 21;16(1):209. doi: 10.1186/s12934-017-0822-6. Feng X et al. Bioresour Technol. 2014 Nov;172:269-275. doi: 10.1016/j.biortech.2014.09.074.). In this study, in addition to obtaining engineered strains that can produce high d-lactic acid, we prefer to investigate the lactic acid tolerance mechanism of Klebsiella pneumoniae and explore essential genes. In order to increase d-lactic acid production, we will further optimize our engineered strains and perform fed-batch fermentations in future studies.
Minor comments:
- What is the mean of high-quality D-lactic acid in the title? I am not able to find the purity of D-lactate in the whole results.
Reply: We sincerely apologize for the unclear description. Actually, the d-lactate dehydrogenase of the K. pneumoniae strain exhibits excellent selectivity, and K. pneumoniae ATCC25955 is able to produce optically pure d-lactate with no detectable l-lactate in culture. (Feng X et al. Bioresour Technol. 2014 Nov;172:269-275. doi: 10.1016/j.biortech.2014.09.074). We have added the explanation in the revised manuscript (Marked-Up Manuscript, lines 45-48).
- Line 16, “productivity” is not accurate, please change to production or yield.
Reply: Thank you for your precious suggestion, we have revised “productivity” to “production” in the revised manuscript (Marked-Up Manuscript, line 16).
- Line 39, delete “can be generated”
Reply: Thank you for your precious suggestion, we have deleted “can be generated” in the revised manuscript (Marked-Up Manuscript, line 39).
- Line 42, it seems the fermentation medium is very complicated and contains yeast extract, why the authors think it is a minimal medium?
Reply: We sincerely apologize for the unclear description. In fact, the fermentation medium we used in this study did not contain yeast extract (Marked-Up Manuscript, lines 109-116). We added 3 g/L yeast extract in the process of ALE to speed up the growth of bacteria (Marked-Up Manuscript, line 157).
- Line 98, 37 ℃
Reply: Thank you for your precious suggestion. We apologize for our issues and we have revised this line in the revised manuscript (Marked-Up Manuscript, line 108).
- Line 114-115, suicide plasmids
Reply: Thank you for this precious suggestion. We apologize for our issues and we have revised this word in the revised manuscript (Marked-Up Manuscript, lines 118).
- “OD600” in the text and figure should be unified.
Reply: Thank you for your precious suggestion. We apologize for our issues and we have revised “OD600” to “OD600” in the revised manuscript (Marked-Up Manuscript, lines 143, 146 and 151).

Reviewer 3 Report
Comments and Suggestions for Authors
The topic of the study fits within the scope of the journal, and the experimental work is organized with general methods. There are still few remaining questions/comments listed below in detail:
Some detailed aspects/questions:
The percentage of the cited literature in the range of “2016 or older” is about 55%. From this perspective, it would be necessary to cite the current literature in order to justify more thoroughly the novelty of the present work and/or further substantiate the proposed progress. The update of the references would have impact on the sections “introduction/state of the art” and “results and discussion”!
In addition, it would be helpful to include a table with already published figures comparing the results here obtained. I imagine there are not that many directly comparable papers (for this particular strain) already published, but ultimately the ALE approach presented here has to compete with ALL other already existing technologies/strains for D-LA production in terms of titer, productivity or other performance parameters.
Do you have any information on the pH drops (please add the desired setpoint in section 2.1?) that occurred in the shake flasks during the 12 h (uncontrolled cultivation)? In general, it is advantageous and more reproducible to conduct such comparative experiments at constant pH in a fully equipped bioreactor. I can imagine that continuous pH control is crucial for process performance. Could you please comment on this!?
Furthermore, is there a specific reason to investigate or optimize this Klebsiella strain, as there are other D-LA producers that reach higher product concentrations and do not form by-products (such as those described in 2.5, where acetate in particular is very high!)? Do you have any information about the (long-term) stability of the strain changes/modification?
Author Response
Response to Reviewer 3:
We thank you for your constructive comments for our study. We have carefully revised the manuscript and uploaded a compared copy of the manuscript (without figures) as a "Marked-Up Manuscript" file (changes in the manuscript are marked in red), and here are point-by-point responses:
- The percentage of the cited literature in the range of “2016 or older” is about 55%. From this perspective, it would be necessary to cite the current literature in order to justify more thoroughly the novelty of the present work and/or further substantiate the proposed progress. The update of the references would have impact on the sections “introduction/state of the art” and “results and discussion”!
Reply: Thank you for your precious suggestion. We fully agree that it should be necessary to cite the current literature to justify more thoroughly the novelty of the present work. As a result, we have updated the references in the revised manuscript, now the range of “2017 or newer” is more than 60%.
- In addition, it would be helpful to include a table with already published figures comparing the results here obtained. I imagine there are not that many directly comparable papers (for this particular strain) already published, but ultimately the ALE approach presented here has to compete with ALL other already existing technologies/strains for D-LA production in terms of titer, productivity or other performance parameters.
Reply: Thank you for your precious suggestion. We fully agree that it is helpful to include a table with already published figures comparing the results of our study. We have added Table 3 to the revised manuscript. Table 3 summarizes the d-lactate fermentation performances from different carbon sources by different strains in the shake flask. we added the paragraph (Marked-Up Manuscript, lines 393-404) as following:
" In addition, diverse carbon sources have been employed in d-lactate fermentation, including glucose, glycerol, sweet potato, sugar beet pulp, and corncob slurry. In recent years, studies about engineered Klebsiella pneumonia, Saccharomyces cerevisiae, Escherichia coli, Lactobacillus coryniformis, Pichia kudriavzevii and some other microorganisms for producing d-lactate have been reported. We have compared the d-lactate fermentation capabilities of different strains using different carbon sources in flask fermentation (Table 3). To our knowledge, the production of 19.56 g/L in this study represents the highest d-lactic acid production in flask shake by K. pneumoniae to date. Notably, d-lactic acid fermentation is mainly cultured in a complex medium containing broth or yeast at a relatively high cost. As a result, the high productivity and optical purity of d-lactic in a low-cost minimal medium in this study indicated that the engineered K. pneumoniae Q5224 is an excellent producer of d-lactate.”
- Do you have any information on the pH drops (please add the desired setpoint in section 2.1?) that occurred in the shake flasks during the 12 h (uncontrolled cultivation)? In general, it is advantageous and more reproducible to conduct such comparative experiments at constant pH in a fully equipped bioreactor. I can imagine that continuous pH control is crucial for process performance. Could you please comment on this!?
Reply: Thank you for your precious suggestion. We sincerely apologize for our deficient information in section 2.1 and we absolutely agree that continuous pH control is advantageous for process performance. However, it is difficult in shake-flask experiments. Indeed, after 12 hours, we measured that the pH had dropped to about 4 (Marked-Up Manuscript, line 131), which would inhibit the growth of the microorganism. To control the pH, we utilize the usual fermentation method, adding NH3·H2O every 12 hours. Therefore, the fermenter with pH control can be employed in future studies for optimizing the production of d-lactic acid.
- Furthermore, is there a specific reason to investigate or optimize this Klebsiella strain, as there are other D-LA producers that reach higher product concentrations and do not form by-products (such as those described in 2.5, where acetate in particular is very high!)? Do you have any information about the (long-term) stability of the strain changes/modification?
Reply: We sincerely apologize for our unclear description. We absolutely agree that there are other d-LA producers that reach higher product concentrations and do not form by-products. In this study, we optimize K. pneumoniae for the following reasons:
- Its rapid growth rate in minimal medium and substrate versatility (including glycerol and monosaccharides) (Marked-Up Manuscript, lines 42-43).
- It is already widely used as a microbial factory for the production of 3-hydroxypropionate, 1,3-propanediol, hyaluronic acid, 2-hydroxyisovalerate, and d-lactic acid (Marked-Up Manuscript, lines 43-45).
- The d-lactic acid dehydrogenase of the pneumoniae strain exhibits excellent selectivity, and K. pneumoniae ATCC25955 is able to produce optically pure d-lactate with no detectable l-lactate in our previous studies (Marked-Up Manuscript, lines 46-48).
- There remains a scarcity of reports on metabolic engineering strategies for d-lactic acid production in pneumoniae. Investigating K. pneumoniae further may reveal novel approaches to enhance d-lactic acid production and broaden its industrial applications (Marked-Up Manuscript, lines 50-53).
- This strain is relatively easy to obtain (lab collection)
Thank you a lot for your valuable advice and we rephrased the introduction in the revised manuscript (Marked-Up Manuscript, lines 43-48).
In addition, the high acetate production could be attributed to the addition of excess glucose in the fermentation medium. In the following studies, we can reduce the amount of glucose supplied to the medium to decrease acetate accumulation. On the other hand, acetate is also a valuable carbon source in microbial fermentation. For example, in our previous study, acetate was used as a sole carbon source for higher Poly(3-hydroxybutyrate) production than glucose (Sun S et al. Front Bioeng Biotechnol. 2020 Jul 23;8:833. doi: 10.3389/fbioe.2020.00833.).
Finally, thank you again for your valuable advice. These engineered strains remain stable under our current experimental conditions. However, we do not yet have information about the (long-term) stability of the strain changes. To boost strain stability, we will construct engineered strains with high d-lactic acid production using the indicated beneficial mutations in future studies.

Round 2
Reviewer 2 Report
Comments and Suggestions for Authors
I believe the authors have done an excellent job responding to the concerns of the reviewers. I support publication.
Reviewer 3 Report
Comments and Suggestions for Authors
Thank you very much for the explanations, answers and additions on the manuscript. There are no further questions or comments from my side!